

**Characterization of the Newly Designed Wall-Free Particle Evaporator**
**(WALL-E) for Online Measurements of Atmospheric Particles**
Linyu Gao[1,†], Imad Zgheib[1,#,†], Evangelos Stergiou[2], Cecilie Carstens[1], Félix Sari Doré[1,‡],
Michel Dupanloup[1], Frederic Bourgain[1], Sébastien Perrier[1], Matthieu Riva[1*]
[1] Univ Lyon, Université Claude Bernard Lyon 1, CNRS, IRCELYON, F-69626, Villeurbanne,
France
[2] Environmental Chemical Processes Laboratory (ECPL), Department of Chemistry, University
of Crete, Voutes Campus, 70013 Heraklion, Greece
[#] now at TOFWERK, 3645 Thun, Switzerland
[‡] now at Department of Chemistry and Molecular Biology, Atmospheric Science, University of
Gothenburg, SE-41390, Gothenburg, Sweden
[†] These authors contributed equally to this work
Corresponding to: matthieu.riva@ircelyon.univ-lyon1.fr



**Abstract**
Organic aerosols (OA) play a critical role in the atmosphere by directly altering human health
and climate. Understanding their formation and evolution as well as their physicochemical
properties requires a detailed characterization of their chemical composition. Despite advanced
analytical techniques developed within the last decades, real-time online measurement of
atmospheric particles remains challenging and suffers from different artifacts. In this work, we
introduce the newly designed wall-free particle evaporator (WALL-E) coupled with a chemical
ionization mass spectrometer (CIMS) using bromide (Br$^-$) as the reagent ion. We
comprehensively evaluate the performance of the WALL-E system, demonstrating its ability to
evaporate particles while maintaining the integrity of the compounds composing the particles
(i.e., minimal thermal decomposition). To demonstrate WALL-E's performance, the
composition of aerosol particles formed from α-pinene ozonolysis in the presence of $SO_2$ is
characterized. In addition, by applying the scan declustering method, we can now provide a
quantification of the different species present in the condensed phase, e.g., $C_{10}H_{16}O_4$ 84 ng m$^-$
$^3$, $C_{19}H_{28}O_7$ 7 ng m$^{-3}$ for a total SOA mass of 1 µg·m$^{-3}$. While dimers exhibit higher sensitivities,
they account for only 14-18% of the total particle masses, which is considerably lower than
their signal fractions (23-29%). This suggests a potential overestimation of the dimer
contributions when relying solely on signal fractions. In addition, volatility analysis using
thermograms reveals a clear relationship between $T_{50}$ and compound saturation vapor pressure
(C$^*$), with lower-volatility species desorbing at higher temperatures. In addition, measured $T_{50}$
for α-pinene-derived SOA products agree well with theoretical volatility estimation models
(e.g., SIMPOL). Overall, this study demonstrates that WALL-E system coupled to a CIMS is a
promising technique for real-time particle characterization (i.e., composition, quantification,
and volatility) of atmospheric aerosols.



## 1 Introduction


Atmospheric organic aerosol (OA) particles play a critical role in the Earth's climate system
and atmospheric processes by affecting the radiative forcing, cloud formation and albedo,
atmospheric chemistry, environmental sustainability, and human health (Fehsenfeld et al., 1992;
Laothawornkitkul et al., 2009; Mellouki et al., 2015; Charnawskas et al., 2017). A large fraction
(20-90 %) of fine particles are comprised of OA (Kanakidou et al., 2005), which are estimated
to have a global source of 150 Tg yr$^{-1}$ (Pai et al., 2020). In the atmosphere, OA are either directly
emitted as particles from, e.g., volcanic eruption and biomass burning, or formed from gas-to-
particle conversion from the oxidation of volatile organic compounds (VOCs). Due to the
complex mixture of diverse organic compounds in OA particles, characterization of their
chemical composition remains challenging, notably at high time-resolution. Such
characterization is crucial for understanding particle formation, growth, aging, as well as their
physicochemical properties in the atmosphere. Therefore, improving online detection
techniques for OA particles is essential.
Currently, online mass spectrometry (MS), such as chemical ionization mass
spectrometer, is one of the key technologies for measuring gaseous oxygenated organic species.
It takes advantage of tracking the evolution of compounds during their formation and phase
partitioning. To retrieve particle-phase composition with a mass spectrometer, condensed
molecules must be converted to gaseous analytes before being ionized. The key point for getting
qualitative real-time particle composition information is the design of the inlet in front of the
MS. The current technologies to achieve this conversion can be broadly classified into two
categories: those requiring pre-collection of particles and those that do not. A thermal
desorption chemical ionization mass spectrometer (TD-CIMS) (Voisin et al., 2003; Smith et al.,
2004; Li et al., 2021) uses a metal filament in an electrostatic precipitator to collect pre-charged
aerosol particles and thereafter thermally evaporate them by pulsing a known current on the



filament. This approach requires both particle deposition and subsequent evaporation. Another
instrument involving particle collection is the filter inlet for gases and aerosols (FIGAERO)
developed by Lopez-Hilfiker et al. (2014). The FIGAERO inlet collects particles onto a PTFE
filter and desorbs them afterward using a heated $N_2$ flow. This approach enables the control of
particle evaporation in a pre-set temperature-ramping program, making it an efficient technique
to obtain particle volatility information (Stark et al., 2017; Bannan et al., 2019; Tikkanen et al.,
2020; Thornton et al., 2020) while reaching very low detection limits (Lopez-Hilfiker et al.,
2014; Thornton et al., 2020). However, since the collection and evaporation of aerosol particles
take several tens of minutes, the time resolution of particle measurements remains low,
especially in places with low particle loading, resulting in semi-online measurements. Because
of particle collection, the FIGAERO is also not interference-free from organic mixtures,
especially with high concentrations of aerosol particles (Bannan et al., 2019), which may affect
the retrieved volatility of single components. Finally, it has been shown that during evaporation,
chemical processes can occur, altering the chemical composition and the information retrieved
(Stark et al., 2017; Schobesberger et al., 2018; Buchholz et al., 2020).

In contrast, there are inlets without the need to pre-concentrate particles. An inlet

designed for chemical analysis of aerosols online (i.e., CHARON) has been developed without
the need to collect particles (Eichler et al., 2015). It consists of a carbon strip denuder for gas-
phase compound removal, an aerodynamic lens for particle collimation, and a thermo-
desorption unit for particle evaporation. The evaporated compounds can be analyzed with a
downstream low-pressure MS. However, due to the strong electric field in the ion drift tube,
the protonation-induced ionic fragmentation of oxygenated organic compounds biases the real
distribution of particle-phase chemical composition (Müller et al., 2017; Li et al., 2022; Peng
et al., 2023). Another technique, using an atmospheric pressure chemical ionization Orbitrap
mass spectrometer (APCI), evaporates aerosol particles in a heated ceramic tube where thermal



decomposition compounds can be observed (Vogel et al., 2016; Zuth et al., 2018). Finally, the
most recently developed vaporization inlet for aerosols (VIA) coupled to a $NO_3$-CIMS (VIA-
$NO_3$-CIMS) allows continuous thermal desorption and online detection of particle-phase highly
oxidized molecules without pre-concentration (Häkkinen et al., 2023). While this new coupling
allows the identification of highly oxygenated organic molecules at atmospheric relevant
particle concentration, the design of the VIA yields subsequent thermal fragmentation when the
analytes interact with the heated walls of the TD unit (Zhao et al., 2023; Zhao et al., 2024b).
To prevent thermal fragmentation, the extractive electrospray ionization time-of-flight
mass spectrometer (EESI-TOF-MS) (Lopez-Hilfiker et al., 2019) was developed as an online
method for particle analysis without the need of thermal desorption. In the EESI-TOF-MS,
sampled particles collide with charged electrospray droplets, and the soluble compounds are
extracted and ionized through adduct formation. However, the quantification of molecules
remains challenging due to the uncertainties in the dependence of instrument sensitivity on
molecular identity (Lopez-Hilfiker et al., 2019; Wang et al., 2021). This method exhibits also
important background due to the low selectivity of the reagent ions (e.g., $Na^+$), making the
identification and the quantification of the compounds of interest challenging (Lee et al., 2020;
Bell et al., 2023).
In this work, a newly designed wall-free particle evaporator (WALL-E) is designed to
achieve real-time measurements of aerosol particles while preventing ionization-induced
fragmentation and minimizing thermal decomposition effects. WALL-E is coupled to a
chemical ionization inlet attached to a CIMS. An extensive characterization of the WALL-E
system is presented here, where its performance is tested as a function of various parameters,
e.g., flow rates and evaporation temperatures. The sensitivity of WALL-E coupled to an
atmospheric pressure CIMS is determined and the system is used to quantitatively retrieve the
composition of particle-phase oxygenated molecules generated from the $O_3$/OH initiated



oxidation of α-pinene in an aerosol flow tube reactor. Finally, by scanning the WALL-E
temperature, volatility information can be extracted that can be inferred from the measured
thermograms. Polyethylene glycol (PEG) is used to evaluate the volatility measurements and
compare our results to the existing techniques.
**2 WALL-E design, Experiments, and Instruments**
**2.1 WALL-E setup and simulations**
The WALL-E system is designed to thermally desorb aerosol particles while minimizing the
analyte wall-interactions with the system, thus limiting fragmentation (Figure 1). The system
enables real-time mass spectrometric analysis of aerosol particles by integrating a series of
carefully designed components, including a gas-phase denuder, a thermal desorber (TD) unit
equipped with a sheath flow, a ceramic spacer for thermal isolation, and a dilution/cooling unit.
Each component is optimized to enhance sample stability and compatibility with different types
of CIMS.

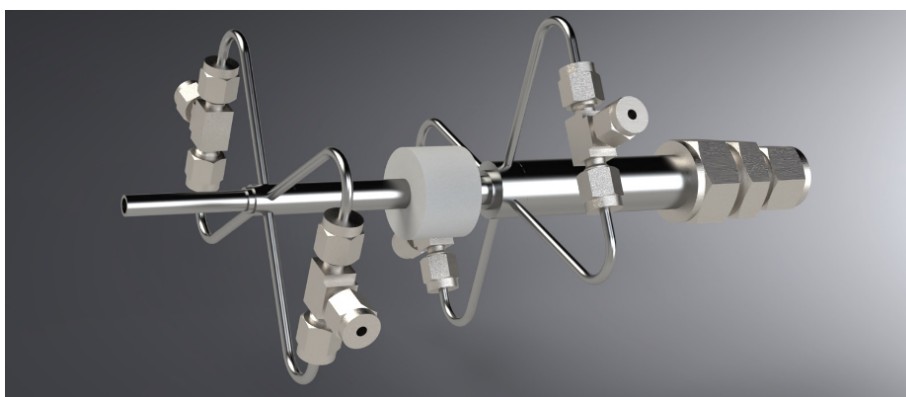


Figure 1. Design of the WALL-E interface.







### 2.1.1 Gas-phase denuder


Like the EESI or the VIA design, the gas-phase denuder is the first stage of the WALL-E
system, designed to selectively remove VOCs and other inorganic gaseous species (e.g., nitric
acid) while allowing aerosol particles to pass through with minimal losses (transmission
efficiency > 90%). This denuder consists of a 10 mm outer diameter, 6 mm inner diameter,
stainless steel tube lined with an activated charcoal honeycomb structure, which provides a high
surface area for adsorption. Maintaining laminar flow is critical for the denuder's efficiency.
With a sample flow rate of 1 SLPM (standard liters per minute), the flow remains laminar with
a Reynolds number of 239, ensuring optimal gas-phase diffusion to the adsorbent walls.

### 2.1.2 Core vaporization unit


The TD is the core of WALL-E, where aerosol particles are converted into gas-phase species
through flash evaporation. It comprises a 4 cm stainless steel tube heated to up to 390°C. As
the sample flow enters the TD region, it merges with a hot nitrogen ($N_2$, up to 390°C) sheath
flow, introduced upstream at a flow rate ranging from 0 to 1 SLPM. The hot sheath-flow serves
multiple purposes: it maintains a laminar flow by preventing turbulences due to T differences
of the sampling flow with the walls of the TD and protects the evaporated analytes from the
heated walls. Hence, the TD's design prioritizes minimal wall interactions to reduce potential
fragmentation of the products, while ensuring that particle evaporation occurs within a well-
defined thermal environment. The stainless-steel tube provides consistent heat transfer along
its length, creating a stable thermal zone for the analytes. The combination of the heated tube
and the hot sheath-flow allows the aerosol particles to rapidly reach the target temperature,
significantly reducing the residence-time required for complete desorption. This rapid
temperature-ramp is essential for achieving flash evaporation, where particles are quickly
vaporized without prolonged exposure to high temperatures that could lead to thermal
decomposition.



Numerical simulations using COMSOL Multiphysics® are used to evaluate and refine

the flow and temperature dynamics within the system, employing models for turbulent flow,
compressible flow, heat transfer, and chemical transport. These simulations show the system's
ability to maintain laminar flow conditions across all units, which is essential to achieve
uniform heating and consistent particle transport. Experimental validation is performed in
conjunction with these simulations, offering valuable insights into both the operational
efficiency and areas for future optimization. The numerical simulations indicate the presence
of uniform heating along the tube length, with thermal stabilization achieved within the first
centimeter (Figure 2, section A). These simulations highlight a well-defined temperature
gradient that optimizes the desorption processes by ensuring a uniform thermal environment
throughout the TD. The combination of controlled flow rates, precise heating, and minimized
wall-interactions results in efficient aerosol evaporation with minimal fragmentation (Figure

2).


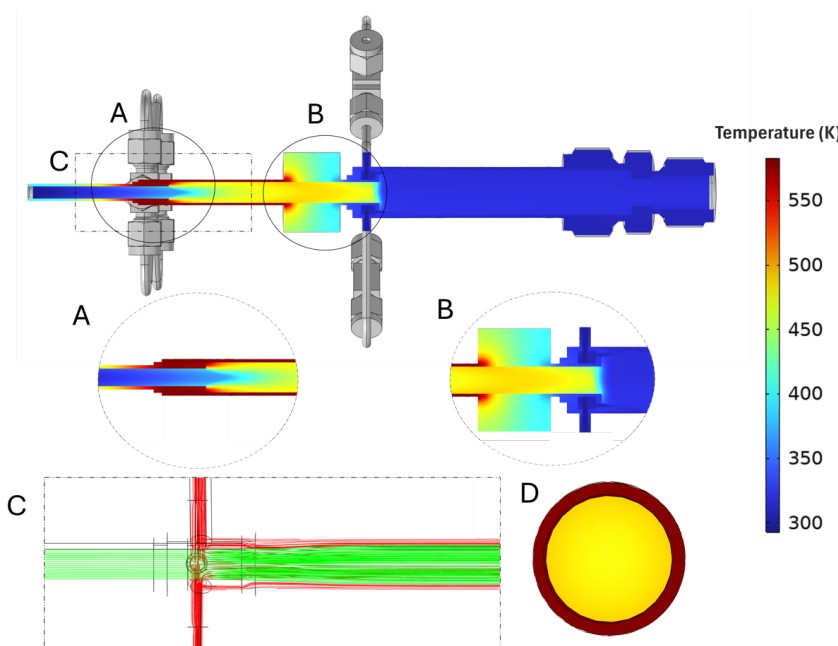





Figure 2: Composite simulation results of the WALL-E system focusing on the thermal
desorber (TD) part. A: Temperature profile in the first hot dilution mixing region, showing
efficient heat transfer and controlled mixing. B: Thermal isolation effectiveness of the ceramic
spacer. C: Streamlines in the sample and TD region, confirming laminar flow. D: Radial
temperature distribution at the TD exit, showing uniform heating across the sample flow.
**2.1.3 Dilution/cooling unit**
After the flash evaporation, a dilution/cooling flow is used by introducing nitrogen (room
temperature) via the second dilution region to prevent re-condensation of vaporized species.
This dilution unit is also a critical step to reduce turbulences caused by temperature gradients
between the TD and the downstream regions as previously observed with the VIA-CIMS
system (Zhao et al., 2024b). A ceramic spacer is positioned between the TD and the cooling
unit to ensure thermal isolation. This spacer, made from Alumina Ceramix $Al_2O_3$ with low
thermal conductivity, minimizes heat transfer between the heated TD and the cooled sample
stream, maintaining distinct thermal zones, as depicted in Figure 2. Simulations confirm the
effectiveness of the ceramic spacer in preserving thermal gradients, greatly reducing unwanted
heat transfer into the cooling unit. To further optimize the cooling unit, two fans are mounted
right after the second dilution region to ensure that the sample reaches room temperature while
preventing turbulence within the CI inlet. This separation is critical to ensure the stability of the
sample as it progresses toward the CI inlet operated at atmospheric pressure. As indicated in
Figure 2, the cooling seems to be achieved within the first centimeter while minimizing
turbulence.
**2.2 Experiments**
**2.2.1 Characterization of the optimal WALL-E setting parameters**



To determine the optimum WALL-E parameters (sampling flow rate, hot sheath-
flow/temperature, and TD temperature), particles are generated using an atomizer (Model 3076,
TSI, Minnesota, USA) with an aqueous solution of citric acid (Sigma-Aldrich, ≥99.5%), d-(+)-
glucose (Alfa Aesar, ≥99%), malonic acid (Sigma-Aldrich, 99%), phthalic acid (Sigma-Aldrich,
≥99.5%), and ammonium sulfate (Sigma-Aldrich, ≥99%). Aerosol particles are dried using a
silica gel dryer, after which the sampling line is divided to provide an aerosol flow to a scanning
mobility particle sizer (SMPS, TSI Incorporated, USA) and to the WALL-E CIMS. The setup
is illustrated in Figure S1 and all parameters tested are summarized in Table S1.
**2.2.2 Sensitivity determination**
The sensitivity of individual compounds is directly proportional to their clustering strength with
the reagent ions (Iyer et al., 2016; Bi et al., 2021), which can be probed by performing a
declustering scanning procedure (Lopez-Hilfiker et al., 2016). Using the setup shown in Figure
S1, single component aerosol particles are generated using an atomizer containing single
component aqueous solutions of the following compounds: 1,5-dihydroxynaphthalene (Sigma-
Aldrich,    ≥97%),    3,4,5-trihydroxybenzaldehyde    (Sigma-Aldrich,    ≥98%),    4-
hydroxyphenylacetic acid (Sigma-Aldrich, ≥98%), ammonium sulfate (Sigma-Aldrich, ≥99%),
citric acid (Sigma-Aldrich, ≥99.5%), d-(+)-glucose (Alfa Aesar, ≥99%), d-mannitol (Sigma-
Aldrich, ≥98%), phthalic acid (Sigma-Aldrich, ≥99.5%), phthalic acid d4 (Sigma-Aldrich,
≥98%), and shikimic acid (Alfa Aesar, ≥98%). The sensitivity, reported in Table S2, for each
type of particle is determined by the linear regression of the mass concentration measured by
the SMPS and the normalized signal intensity of the analyte clustered with Br⁻ (i.e., [M-Br⁻])
detected by the WALL-E-CIMS (Figure S2). To further assess the correlation between the
sensitivity and the binding energy of the detected ion adducts, voltage scanning (i.e., increasing
the voltage difference between two ion optics) is performed to determine the half-signal



maximum intensity (Lopez-Hilfiker et al., 2016; Riva et al., 2020) for the generated single-
component aerosol particles.

### 2.2.3 SOA particle generation

To further examine the performance of the WALL-E system, SOA are generated in an 18-liter
Pyrex glass aerosol flow tube reactor (12 cm i.d. × 158 cm length) from the $O_3$/OH initiated
oxidation of α-pinene in the presence of $SO_2$ at room temperature and atmospheric pressure
(Stein and Scott, 1994). Ozone ($O_3$) is stably generated by passing a flow of 0.6 SLPM of
synthetic air after exposure to a UV lamp (Ozone Generator Model 610, Jelight Company, Inc,
Irvine, USA). $SO_2$ is injected from a commercial cylinder (500 ppm, AIR PRODUCTS Inc.).
α-Pinene was introduced from a pressurized cylinder (40 ppm in nitrogen). $O_3$, $SO_2$, and α-
pinene are continuously injected into the aerosol flow tube reactor to generate a total aerosol
mass ranging from 1.0 to 15.6 µg·m$^{-3}$. The concentrations of reactants are summarized in Table
S3. A mixture of nitrogen and oxygen (total flow 4 SLPM) is used as a carrier gas, providing a
reaction time of ~4.5 minutes. Before injecting α-pinene, background measurements are
obtained.

### 2.2.4 Thermograms and $T_{max}$ determination

The volatility distribution of aerosol particles is investigated using thermograms obtained with
the WALL-E system. Polyethylene glycol (PEG-400) aerosols are produced by atomizing
aqueous solutions as performed in previous studies (Ylisirniö et al., 2021; Zhao et al., 2024b)
and sampled by the WALL-E system. The temperature of the WALL-E TD is gradually
increased from 30°C to 390°C in 30°C increments every 10 minutes. The PEG standards,
ranging from PEG-6 to PEG-17, have been chosen to represent a wide range of molecular
weights and volatilities (Krieger et al., 2018).



**2.3 Instrumentation**

Within these experiments, WALL-E is associated with an atmospheric pressure CI inlet (Riva et al., 2019a; Riva et al., 2020) coupled to an Orbitrap (Q-Exactive, Thermo Fisher Scientific) utilizing bromide ions ($Br^-$) as the reagent ion. $Br^-$ is generated from dibromomethane (Sigma-Aldrich, 99%) continuously flushed by 2 standard cubic centimeters per minute (sccm) of pure $N_2$, and subsequently ionized with a soft X-ray photoionizer (Hamamatsu, L9491). The sheath and the total flows are 24 SLPM and 33.5 SLPM, respectively. The Orbitrap is operated with an automatic gain control (AGC target) of $1 \times 10^6$ charges, an S-lens radio frequency level of 50, a maximum injection time of 1000 ms, 10 microscans, and a capillary temperature set to 150 °C. The mass resolution is 140,000 (at m/Q 200). Orbitool 2.2.4  (Cai et al., 2021) is used for analyzing the data. The data are pre-averaged to 1 minute. Signals are background subtracted and normalized by the signal intensity of $Br^-$ (m/Q 79). To obtain an accurate concentration of compounds present in low abundance, a linearity correction (Riva et al., 2020) is applied to all measured signals (Figure S3).

The mass concentration of particles is retrieved using a scanning mobility particle sizer (SMPS) utilizing a differential mobility analyzer (DMA; 3081, TSI Inc.) connected to a CPC (3772, TSI Inc.), by applying an assumed particle density (1.45 $g/cm^3$) (Kim et al., 2010; Shilling et al., 2009) for aerosol particles generated from the $O_3$/OH initiated oxidation of α-pinene in the presence of $SO_2$. An impactor (0.071 m) is used, and the sampling flow rate is 1 SLPM with a sheath flow of 10 SLPM.

**3 Results and Discussion**

**3.1 Optimal setting parameters of WALL-E**

As described in section 2.1, WALL-E involves multiple operational parameters, including the sample flow rate (SF), the hot sheath flow rate (HF) and its temperature upstream of the TD,



270 the dilution flow in the cooling region, and the TD temperature region. The cooling flow is held

271 constant at 10 SLPM to minimize turbulence in the CI inlet operated at atmospheric pressure.

272 To determine the optimized parameters, various combinations of SF and HF flowrates as well

273 as HF and TD temperatures are tested on aerosol particles generated from an atomized water

274 solution containing a mix of standards. All data are corrected for their respective dilution

275 factors. Figure 3 shows the summed signal of all evaporated products (i.e., phtalic acid, citric

276 acid, malonic acid, glucose, sulfuric acid) under these different conditions, normalized to the

277 maximum value.

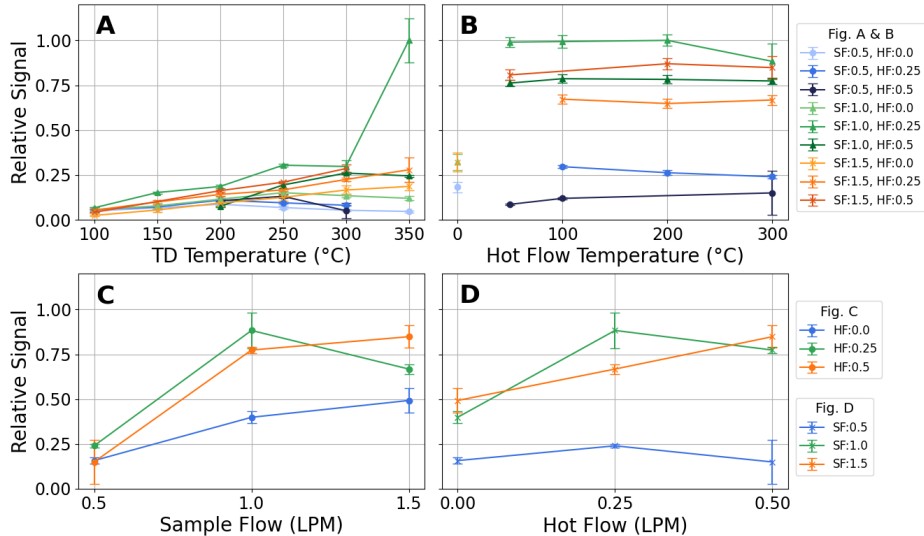

278 Figure 3: (A) Relative signal at a fixed hot sheath-flow (HF) temperature of 300 °C versus TD

279 temperature for various combinations of sample flow rate (SF) and HF. (B) Relative signal at a

280 fixed TD temperature of 300 °C versus HF temperature. (C) Relative signal versus SF at

281 different HF settings. (D) Relative signal versus HF at different SF values. Both (C) and (D)

282 are at a fixed TD and HF temperature of 300 °C. SF is the sample flow rate delivered to the TD,

283 while HF is the heated dilution flow added upstream of the ambient dilution stage. The second,

284 ambient dilution flow and the downstream cooling flow remain fixed to minimize turbulence





in the CIMS. Error bars represent standard deviations based on repeated measurements at the
same conditions, and all values are corrected for their respective dilution factors.
As depicted in Figure 3A, it is evident that a SF of $1.0\,L\,min^{-1}$ consistently results in higher
signals than $0.5\,L\,min^{-1}$ across all TD temperatures. A SF of $1.5\,L\,min^{-1}$ also performs
reasonably well, whereas the $0.5\,L\,min^{-1}$ configurations exhibit significantly lower signals. A
similar trend is observed in Figure 3B, where the relative signal is plotted against the HF
temperature. The results indicate that efficient evaporation occurs at a TD temperature of
around 300-350°C, while variations in the HF temperature have a less important effect.
However, the presence of a HF (either 0.25 or $0.5\,L\,min^{-1}$) at an SF of 1 or $1.5\,L\,min^{-1}$ enhances
evaporation efficiency and flow stability. This influence of flow conditions is more distinctly
observed in Figures 3C and 3D and further illustrates the impact of SF and HF on the relative
signal. Figure 3C presents the relative signal as a function of SF for different HF settings,
demonstrating that a SF of $1\,L\,min^{-1}$ and a HF of $0.25\,L\,min^{-1}$ yield higher signal intensities
compared to other conditions. Similarly, Figure 3D shows the relative signal plotted against HF
at different SF values, reinforcing this observation.

A key consideration when selecting optimal SF and HF conditions is to keep

fragmentations minimal. Excessive heating in the TD can enhance fragmentation, potentially
resulting in thermal decomposition of the analytes. To assess this, we evaluate the thermal
decomposition of citric acid, which is a known analyte to decompose within TD inlets (Yang
et al., 2023). As shown in Figure S4, the thermal fragmentation of citric acid using the
conditions (i.e., SF of $1\,L\,min^{-1}$ and a HF of $0.25\,L\,min^{-1}$) described above remains negligible,
with up to 2‰ of the total signal attributed to fragment ions at the hottest temperature (i.e.,
390°C). This confirms the minimal fragmentation while keeping an effective evaporation.



### 3.2 Characterization of SOA derived from α-pinene ozonolysis

### 3.2.1 Raw mass spectra

To underline the performance of WALL-E in measuring and quantifying a complex mixture of OA, the characterization of SOA generated from the oxidation of α-pinene in the presence of $SO_2$ is used as an example. Figures 4 and S5 present the mass spectra of α-pinene-derived SOA with particle mass concentrations ranging from 1.0 to 15.6 µg m$^{-3}$. These mass concentrations cover atmospherically relevant ranges observed in remote and rural places (Jimenez et al., 2009), which is achieved when using the optimal setting parameters determined in section 3.1. A total of 146 organic monomers ($C_{1-10}H_yO_zS_0$) and 206 dimers ($C_{11-20}H_yO_zS_0$) are identified, respectively contributing 67-74 % and 29-23 % to the total signal intensity. The dominant ions identified in the monomer region are assigned to $C_{8-10}H_{12-18}O_{3-7}$ compounds with high-resolution peak fitting in the range of m/Q at 249-343 Th, and $C_{17-20}H_{26-34}O_{5-11}$ compounds in the dimeric range between 405-525 Th. Among the monomers, $C_{10}H_{14,16}O_{3-7}$ are the most abundant ones, followed by $C_9H_{14}O_{3-7}$. The product distribution measured is consistent with previous studies (Zhang et al., 2015; Kahnt et al., 2018; Zhao et al., 2023; Zhao et al., 2024a). It should be mentioned that $C_8H_{12}O_4$ is the most abundant individual compound, which could be norpinic acid or terpenylic acid identified by previous studies (Zhang et al., 2015; Du et al., 2022; Witkowski et al., 2023). It has an unexpected high background of ~0.04 ncps when there is no VOC injected. Desorption from the walls of the aerosol flow tube reactor when aerosol particles are produced can explain the presence of $C_8H_{12}O_4$ as reported in previous laboratory studies (Riva et al., 2019b; Wong et al., 2022).

Without sensitivity correction, compounds are significantly detected at a total signal of 0.04 ncps when the particle mass is $1.0 \pm 0.1$ µg m$^{-3}$. As particle mass concentration increases, the normalized signals of particle-phase products also rise, reaching 0.8 ncps at $15.7 \pm 1.2$ µg m$^{-3}$. The detected products exhibit a strong linear response to particle mass concentration, as



shown in Figure S6 illustrating the suitability of the WALL-E system at measuring aerosol
particles under atmospheric relevant conditions.

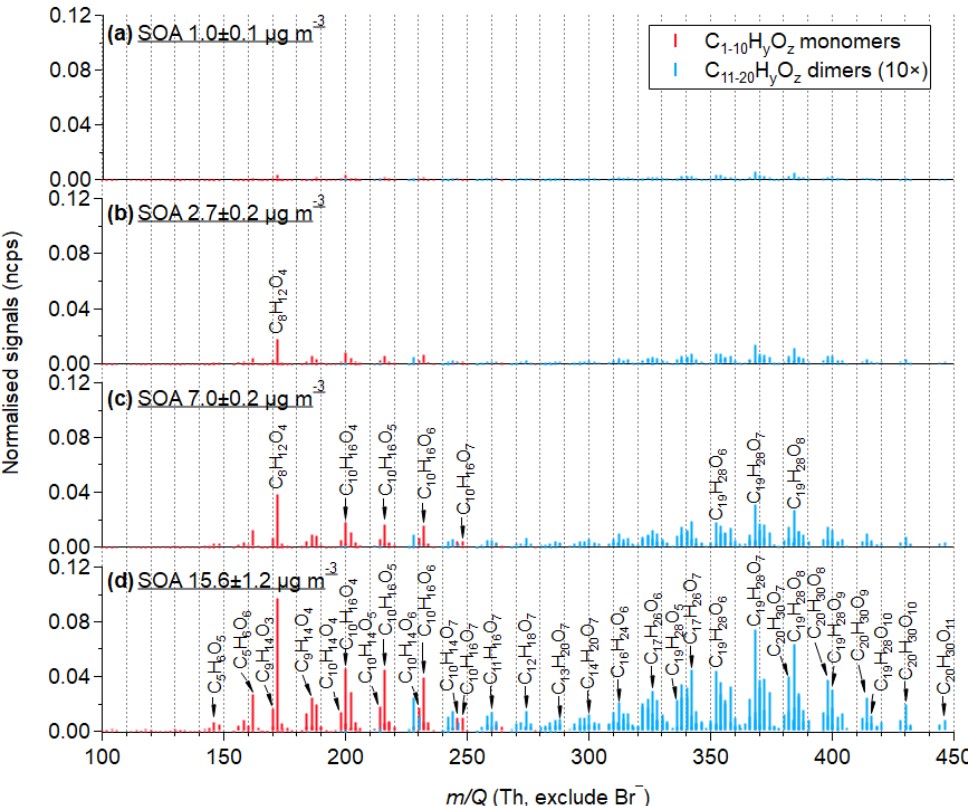


Figure 4. Mass spectra of particle-phase organic compounds formed from the oxidation of α-
pinene under varying mass particle concentrations (1.0-15.6 µg·m$^{-3}$). Compounds are
evaporated and detected by the WALL-E CIMS with HF and TD temperatures of 300 ⁰C, SF of
1 SLPM, HF of 0.25 SLPM, and a cold dilution flow of 10 SLPM. Signals are normalized to
Br$^-$ signal intensity. Red and blue refer to monomers (C$_{1-10}$H$_y$O$_x$) and dimers (C$_{11-20}$H$_y$O$_x$),
respectively. The normalized signals of dimeric compounds are multiplied by a factor of 10.




### 3.2.2 Sensitivity determination on SOA molecules

The sensitivity of standard compounds, which is determined by the linear regression between the normalized signals detected by WALL-E and the particle mass concentrations (Figure S2), is summarized in Table S2. The $r^2$-values of the fitting for most standard compounds are good (0.93-0.99), while 1,5-dihydroxy naphthalene has the lowest $r^2$-value of 0.85. By utilizing the in-source collision ion dissociation feature (Riva et al., 2019a), which corresponds to an increase in the DC offset voltages between two ion optics within the flatapole, the binding energy of the $[M-Br^-]$ adducts of the different compounds can be probed (Figure S7). Their $dV_{50}$ values broadly range from 5.1 to 20.2 Volts, indicating differences in binding energies and varying clustering strengths to $Br^-$. Consequently, the correlation between the sensitivity and the $dV_{50}$ values obtained from the standard compounds is fitted by a non-linear sigmoidal function (Figure S8), which is consistent with prior studies using the same approach to quantify gaseous species (Lopez-Hilfiker et al., 2016; Iyer et al., 2016; Zaytsev et al., 2019; Xu et al., 2022). This calibration curve provides an estimation of the system sensitivity based on experimentally obtained $dV_{50}$ cluster values from declustering scans with increasing energy (Lopez-Hilfiker et al., 2019). Using this method, raw MS signal intensities can be converted into quantified amounts, reducing the need for compound-specific calibration when authentic standards are not available. This method enables semi-quantification across a wide variety of molecules.

By applying the correlation between sensitivity and $dV_{50}$ obtained from the different standard compounds, and using the $dV_{50}$ values determined for individual α-pinene-derived SOA products, every oxidation compound can be quantified. As an example, Figure S9 presents the declustering profiles of $C_{10}$ monomers and $C_{19-20}$ dimers from SOA. Consistent with previous studies (Riva et al., 2019a), more oxidized compounds exhibit stronger binding energies, resulting in higher $dV_{50}$ values. As shown in Figure S10, the corresponding sensitivity





of α-pinene-derived SOA compounds generally increases with molecular mass and reaches a
plateau corresponding to the maximum sensitivity (i.e., collision limit), with an upper limit of
sensitivity of 0.08 ncps·per·μg·m$^{-3}$.

### 3.2.3 Sensitivity-corrected chemical composition of SOA particles

By assessing the sensitivity of individual α-pinene-derived SOA compounds, the mass
concentrations of all identified particle-phase oxidation products can be estimated (Figure S11).
Using the correlation between sensitivity and $dV_{50}$ based on 10 standard compounds, the total
particle mass concentration is estimated to be 27.1 μg·m$^{-3}$, 74% higher than the SMPS
measurements (15.6 μg·m$^{-3}$), assuming spherical particles with an aerosol density of 1.45 g/m$^3$
(Kim et al., 2010; Shilling et al., 2009). By excluding the two outliers (i.e., shikimic acid and
glucose), the estimate of total particle mass concentration (with 36% overestimation) is closer
to the total particle mass concentration measured by the SMPS as depicted in Figure 5. As
discussed in previous studies, selected standard compounds might induce uncertainty in the
sensitivity estimations (Zaytsev et al., 2019; Bi et al., 2021; Song et al., 2024). Notably, the
presence of different isomers can yield substantial uncertainties, especially when their
sensitivity may vary by an order of magnitude (e.g., Lee et al., 2014). It should also be
mentioned that the total mass concentration determined by the SMPS is prone to uncertainties
(Wilson et al., 2015; Bell et al., 2023). Using the $dV_{50}$ method presented here and considering
a total SOA mass of 15.1 μg m$^{-3}$, the mass concentration of $H_2SO_4$ is estimated at 8.7 μg m$^{-3}$
(using 10-compound fitting) and 6.4 μg m$^{-3}$ (using 8-compound fitting). Comparing with the
direct calibration discussed above (i.e., Table S2), a good agreement is retrieved (7%
underestimation and 25% overestimation for the 10 and 8-compounds fit, respectively)
underlining the benefit of using this approach to obtain the concentrations of organic and
inorganic present in the particles. Overall, a linear relationship exists with the total SOA mass
concentrations in the range of 1.0-15.6 μg·m$^{-3}$ (Figure 5) and exhibits an overall good



agreement between WALL-E and SMPS measurements (i.e., 20-30%). Hence, this method can
provide a deeper understanding of aerosol composition and evolution. For example, due to their
general higher sensitivities, dimers contribute only 14-18% to the total mass (Figures 5, S10),
much less than their fractions based on signal intensity (23-29%). This suggests that the dimer
contributions to particles may be overestimated when solely based on signal intensity.

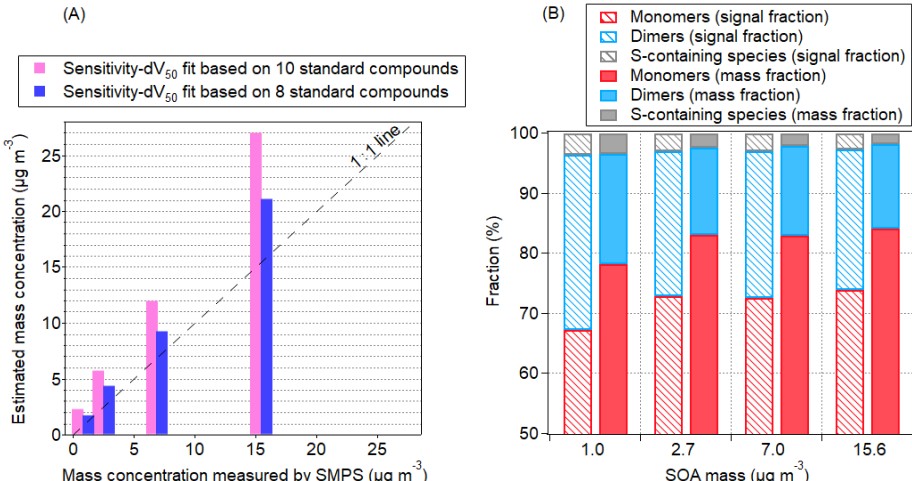


Figure 5. (A) The estimated mass concentration measured by Wall-E as a function of SOA mass
concentration measured by the SMPS. (B) Fractions of monomers, dimers and sulphur-
containing compounds in the SOA particles formed from $O_3$/OH initiated oxidation of α-pinene
in the presence of $SO_2$. The calculation of fractions is weighted by normalized signals (dashed
bars) and mass concentrations (solid bars), respectively.
**3.3 Assessing particle molecular volatility**
**3.3.1 Thermograms and $T_{max}$ determination**
The determination of volatility represents one of the greatest analytical challenges when
characterizing aerosol particles, as it depends on multiple factors, including molecular
composition, intermolecular interactions, and experimental conditions (Compernolle et al.,
2011). Various experimental and theoretical techniques have been developed over the last





decades to retrieve volatility information, each with advantages and limitations. Filter-based
techniques and thermal desorbers provide direct measurements of desorption profiles, while
theoretical models, such as SIMPOL and COSMO-RS, offer predictive estimates of volatility
based on molecular structure. However, volatility determination remains not well constrained,
particularly for highly oxygenated and multifunctional compounds, where structural
differences, such as isomerism, can significantly influence volatility (Lee et al., 2014; Bannan
et al., 2019).
While FIGAERO and VIA have been widely used, their design constraints introduce
inherent limitations. FIGAERO, being a filter-based technique, can introduce artifacts such as
recondensation, analyte interactions, and fragmentation. The prolonged residence time on the
filter may also lead to early desorption of volatile species or chemical reactions between co-
deposited compounds, impacting the accuracy of volatility estimates (Stark et al., 2017;
Schobesberger et al., 2018; Buchholz et al., 2020). VIA thermograms, on the other hand, show
evidence of fragmentation and thermal decomposition at high temperatures (Zhao et al., 2024b).
WALL-E introduces a new approach, optimizing the balance between thermal residence time
and evaporation efficiency, allowing for precise volatility determination with reduced wall
interactions.
**3.3.2 From $T_{max}$ to $T_{50}$**
The $T_{max}$ values, which correspond to the peak desorption temperature, represent the
temperature at which the maximum desorption rate occurs. Traditionally, $T_{max}$ has been used to
estimate volatility from thermograms. This approach is well-suited for FIGAERO and VIA,
where the thermograms typically exhibit a near-Gaussian profile, with a clear peak followed by
a steep signal decline due to decomposition and fragmentation. However, in the case of WALL-
E, where thermograms more closely follow a sigmoidal desorption trend, $T_{max}$ determination





becomes less straightforward. At high temperatures, the signal does not drop sharply but instead
asymptotically reaches a plateau, making $T_{max}$ highly sensitive to noise and minor variations in
the upper temperature range.

To overcome this limitation, we propose using $T_{50}$, the temperature at which the signal

reaches 50% of its maximum as a more robust volatility metric. Since $T_{50}$ is located in the
steepest region of the sigmoid fit, it is significantly less affected by noise, baseline shifts, or
small variations in signal intensity. Unlike $T_{max}$, which depends on the choice of an arbitrary
threshold (e.g., 99.5% or 99.9% of the signal maximum), $T_{50}$ is a main feature of the sigmoid
function, making it a more consistent and reproducible parameter for comparing volatility
trends in the case of WALL-E. To accurately determine $T_{50}$, the thermograms are smoothed and
fitted using a sigmoid function. This approach minimizes the influence of temperature ramping
increments and instrumental noise. Figure 6 displays thermograms for PEG standards,
illustrating that less volatile compounds require higher temperatures for complete evaporation,
whereas more volatile compounds evaporate at lower temperatures.

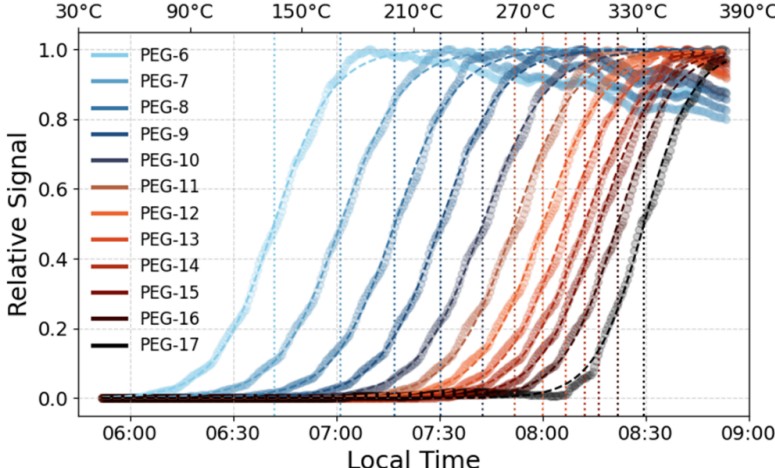

Figure 6: Thermograms of PEG standards obtained using WALL-E. The desorption profiles
illustrate the relative signal intensity as a function of temperature.





The use of a sigmoid fit for the WALL-E data is justified by the observed thermograms, which
closely resemble this behaviour, with only a moderate signal decrease at the highest tested
temperatures. For comparison purposes with other particle evaporators, $T_{max}$ is used with a
signal representation of 98%. Figure S12 shows the effect of varying this signal percentage
threshold. By varying between 99.5% and 98%, a difference of about 10% in the $T_{max}$ value is
observed, which makes it challenging to select an appropriate threshold. In addition, at higher
temperatures, the fit for less volatile PEGs can result in more errors due to fewer data points.
Taking a threshold around where the signal is half always results in more accurate results even
with 10% variations in the threshold. The signals of more volatile compounds, such as PEG-6
and PEG-7, with $T_{max}$ values of 203.6°C and 246.3°C, decrease by only 11.2% and 18.9%
respectively, even after a temperature increase of over 150°C, which represents a significant
improvement compared to other online TD techniques. At this point, PEG-16 and PEG-17 reach
their $T_{max}$ (Table S4). This demonstrates WALL-E's ability to maintain signal integrity across
a wide volatility range, ensuring reliable thermal desorption without excessive signal loss.
**3.3.3 Temperature Correction and Comparison with VIA and FIGAERO**
The temperatures applied during thermal desorption in WALL-E do not directly reflect the
actual gas-phase temperatures experienced by the desorbed compounds. Due to thermal lag and
heat transfer dynamics, a correction factor was derived based on the COMSOL CFD
simulations. The internal gas temperature at the core of the TD was found to follow a linear
relationship with the set temperature (Figure S13). This correction ensures that the reported
$T_{max}$ values accurately represent the true volatility behaviour of the analytes.  The corrected and
uncorrected $T_{max}$ values for each PEG compound are plotted against their molecular masses in
Figure S14. The corrected values align well with those reported for VIA and FIGAERO,
maintaining a consistent trend across different molecular weights. In contrast, the uncorrected
$T_{max}$ values are systematically higher, displaying a different slope compared to the other two

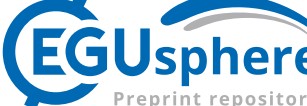



systems. This highlights the necessity of applying the correction factor to ensure accurate
volatility determination.

$T_{max}$ values from WALL-E and VIA are higher compared to FIGAERO, which can be

attributed to the short residence time in the TD region (Zhao et al., 2024b). Additionally,
WALL-E exhibits lower $T_{max}$ values than VIA, likely due to the introduction of heated dilution
flow into the sample stream. This promotes flash evaporation, causing compounds to desorb
before reaching full thermal equilibrium, ultimately shifting $T_{max}$ to lower values while
maintaining the expected volatility trend. As illustrated in Figure S15, the residence time in the
TD region further influences $T_{max}$ values. Longer residence times allow for gradual heating and
equilibration, leading to higher $T_{max}$ values. In contrast, shorter residence times accelerate
desorption, resulting in lower $T_{max}$ values due to insufficient thermal equilibration.
**3.3.4 Volatility Estimation from $T_{50}$**
As previously explained, the use of $T_{50}$ would provide a more reliable estimation of the volatility,
which is typically inferred using the relationship between $T_{50}$ and the saturation concentration
(C*) as discussed in prior studies (Ylisirniö et al., 2021). The wide range of $T_{50}$ values observed
for PEG standards underscores the broad applicability of WALL-E for volatility
characterization across diverse classes of compounds. To derive volatility estimates, we apply
the parameterization method proposed previously (Krieger et al., 2018; Ylisirniö et al., 2021),
which utilizes measured vapor pressures for PEG-5 to PEG-8 and extrapolates the trend for
higher molecular weights. Additionally, an alternative approach using the parameterization (Li
et al., 2016) is considered. While both methods produce similar trends for lower-mass PEGs,
they diverge significantly at higher masses, reflecting the inherent uncertainties in extrapolating
volatility predictions. Given these discrepancies, we define a volatility range (Figure 7) that



encompasses both parameterizations, providing a more robust estimation framework until
additional direct vapor pressure measurements become available.

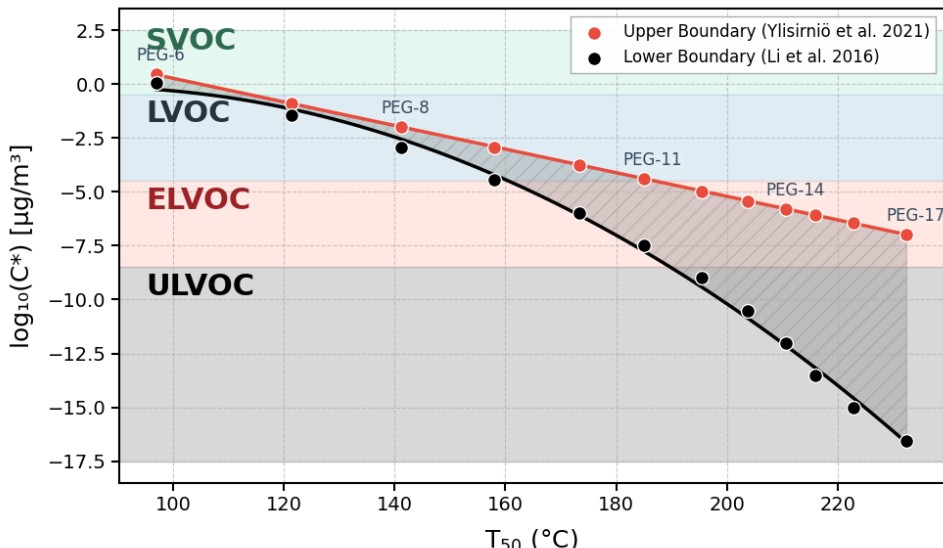

Figure 7: Volatility estimation using measured $T_{50}$ values, ranging from SVOC to ULVOC.
To assess the applicability of $T_{50}$-based volatility determination, we apply this approach to α-
pinene derived SOA compounds, specifically $C_{10}H_{16}O_{6-9}$. The $T_{50}$ values of these SOA species
are determined using multiple ramp-up speeds, ensuring reproducibility across different heating
rates. Additionally, a faster temperature ramp-down from 390°C to 30°C is used as an
independent validation method. As shown in Figure 8, the $T_{50}$ estimated using WALL-E for the
$C_{10}H_{16}O_{6-9}$ from the heating and cooling phases (i.e., temperature ramping up and down,
respectively) exhibit a very good agreement confirming the robustness of the system.

Determined $T_{50}$ values are further compared with theoretical volatility estimation

models, specifically COSMO-RS and SIMPOL, as presented previously (Kurtén et al., 2016;
Peräkylä et al., 2020). Our measured $T_{50}$ values for the SOA compounds fall well within the
SIMPOL-predicted region, confirming that the volatility estimates obtained using WALL-E are
consistent with theoretical predictions. As mentioned earlier, without a pre-separation method,



the presence of isomers can alter the quantification of the compounds of interest, which is also
the case for volatility estimation. As previously discussed by Kurtén et al. (2016) and Peräkylä
et al. (2020), isomeric structures can exhibit significantly different volatilities, reinforcing the
need to account for molecular configurations beyond elemental composition when interpreting
volatility trends.

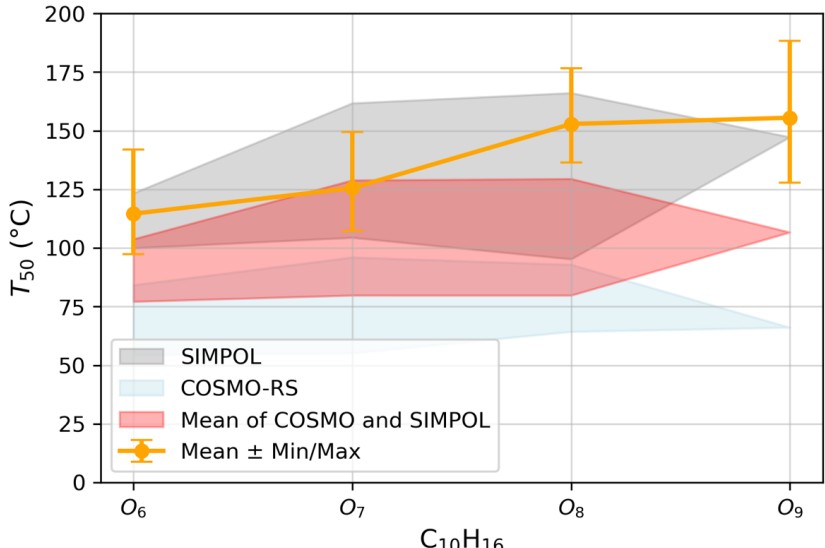

Figure 8: Comparison of estimated $T_{50}$ values for α-pinene derived SOA compounds,
specifically $C_{10}H_{16}O_{6-9}$ with volatility predictions from COSMO-RS and SIMPOL models. The
measured $T_{50}$ values (orange) are shown with their min/max range, while the shaded regions
represent model predictions taking into consideration the isomerisation.
**4 Conclusions**
Within this work we present a new analytical method (i.e., WALL-E) to retrieve the chemical
composition of atmospheric particles in real time. Coupled with a CIMS using $Br^-$ ion chemistry
as the reagent ion, WALL-E is comprehensively characterized, achieving efficient particle
evaporation with maximum evaporation efficiency while exhibiting minimal thermal
decomposition across a range of operational settings. The characterization of SOA produced



from the $O_3/OH$ initiated oxidation of α-pinene in the presence of $SO_2$ further validates WALL-
E's performance in resolving and quantifying complex organic aerosol mixtures. It successfully
identifies a broad range of monomeric ($C_{1-10}$) and dimeric ($C_{11-20}$) compounds. By utilizing the
in-source collision ion dissociation feature, the sensitivity of the analytical method is realized
using a wide variety of authentic standards used to determine the correlation between binding
energy and sensitivity. By using this function and performing declustering procedures,
individual α-pinene derived SOA compounds are quantified at concentration as low as 10 pg·m$^-$
$^3$ (for a total SOA mass of 1 µg·m$^{-3}$). The total estimated SOA mass concentrations is in good
agreement with particle concentration measurements obtained by an SMPS, demonstrating the
benefit of this approach. Notably, the mass contribution of dimeric compounds is determined,
which reveals that they account for only 14-18% of total particle mass, which is notably lower
than their fractions (23-29%) based on signal intensity. Finally, the volatility assessment using
thermogram analysis demonstrates WALL-E's capability to retrieve $T_{50}$ values with high
precision, aligning well with predicted SIMPOL volatility. Future studies should focus on
systematically characterizing SOA volatilities across a broader range of precursor compounds
and oxidation conditions, leveraging complementary mass spectrometry and computational
modelling techniques to refine volatility estimation approaches.
Overall, WALL-E represents a useful and promising tool for atmospheric research, bridging
important gaps in real-time aerosol characterization, quantification of chemical composition of
complex particulate organic mixtures, and volatility assessment. It improves the time resolution
and minimizes measurement artifacts notably due to thermal fragmentation, providing a new
technique for investigating the real-time changes in the formation and growth of atmospheric
particles for laboratory and field observations.
**Data availability**
Data of all figures and tables are available on request from the corresponding author.



**Author contributions**


MR, SP, IZ, MD, and FB designed and built the WALL-E. LG, IZ, ES, CC, and FSD conducted
the experiments. LG, IZ, ES, and FSD analyzed the data. IZ did the simulation. LG and IZ
prepared the paper with contributions from all co-authors.
**Competing interests**
The authors declare no competing financial interest.
**Acknowledgement**
The authors thank Dr. Siegfried Schobesberger for discussing and sharing the FIGAERO
measurements and Dr. Georgios Gkatzelis for discussing the declustering scan.
**Financial support**
This work is funded by the European Research Council Grant (ERC-StG MAARvEL; 423 No.
852161)**.** IZ acknowledges funding from the CLOUD-DOC project (Grant Agreement No.
101073026) under the HORIZON-MSCA-2021-DN-01 program.



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
