# Peer review of "(WALL-E) for Online Measurements of Atmospheric Particles"

_EGUsphere, 2025_

## Author Comment (AC1)

The authors thank the reviewer for the careful review of our manuscript and the helpful comments and suggestions. All the comments (in black) are addressed point by point, with our response in blue, and the corresponding revisions to the manuscript in red.

**Review # 2**

Gao et al. presented a newly designed Wall-Free Particle Evaporator (WALL-E) inlet for online chemical measurements of atmospheric particles using a chemical ionization mass spectrometer (CIMS). The authors claim WALL-E can efficiently evaporate organic particles with minimum thermal decomposition, which can be a good technique for real-time particle characterization. However, some sections are not very clear or convincing, which makes me unsure about the reliability of WALL-E. Please see my comments below for more details. Thus, I suggest major revision before publishing this paper.

General comments:

1. I have several questions about the thermal profile of WALL-E system:

1). The temperature profile in Figure 2 is based on the COMSOL simulation. Although COMSOL is a powerful tool to understand the system's thermodynamics, which is good for system design, the simulation results might not match the real results. Please comment on this by discussing the difference between the simulation results and real conditions. I suggest adding some temperature measurements to confirm the temperature profile. You can add a thermocouple at the center of the tubes. This is also important for your temperature correction.

We conducted temperature measurements using a thermocouple placed at the center of the tube to validate the COMSOL simulation. As shown in the updated Figure S13, the measured and simulated temperature profiles are in good agreement. To account for any remaining differences, we used the mean of the two profiles as a correction function for temperature calibration.

[Figure]

**Figure S13.** WALL-E temperature correction. $T_{set}$ represents the set temperature for both hot sheath flow and TD and $T_{sim}$ and $T_{meas}$ represent the simulated and measured temperatures.

2). Are there any reasons that you don't put any insulation outside the WALL-E system to prevent the thermal exchange with the ambient?

We confirm that insulation is installed around every heated part of the system to minimize heat exchange with the ambient environment and ensure stable thermal conditions during operation. This is not represented in Figure 1 to ease the visualization of the interface.

Lines 164 – 165: *"All heated regions are insulated to minimize heat exchange with the ambient environment."*

3). In the Dilution/cooling unit, I do not understand how this can prevent the re-condensation of vaporized species. I expect less volatile species could condense on the wall. I think you need to do more tests with different ambient relevant species to investigate this.

The cooling and dilution units are specifically designed to minimize re-condensation by introducing a sufficient flow of nitrogen and ensuring fast transfer to the mass spectrometer. Based on the WALL-E configuration, the residence time in the cooling section is very short, estimated between 0.1 and 0.15 seconds, which effectively prevents re-condensation under the tested conditions. We would like to point out that a similar approach is used on the VIA-NO$_3$ interface (Zhao et al., 2024). While we cannot certify the lack of nucleation/recondensation within WALL-E, this has not been observed in the VIA-NO$_3$ interface, while the smaller dilution ratio (3:7) and the longer residence time in the cooling part are more favorable to observe the recondensation of highly oxidized species within the VIA interface.

4). Also, the thermal decomposition was only checked for one species. I suggest adding more standards to validate that, especially those relevant to ambient organics.

In this study, citric acid was selected as a representative compound to assess thermal decomposition because it is a widely recognized benchmark used in previous studies (e.g., Yang et al. (2021)). Its well-characterized and rich decomposition behavior makes it particularly suitable for evaluating the thermal stability of organic compounds in thermal desorption systems, as shown in previous studies (e.g., FIGAERO). As a result, citric acid was used to compare the performance of WALL-E with other systems. Investigating the potential thermal decomposition of a variety of compounds would be the scope of a dedicated study that has been performed by Yang et al. (2021). and is outside the scope of this work.

2. I also have a hard time following the SOA sections.

1). Why do you add SO2 in the SOA generation? It seems very unusual in the literature. You will generate organic sulfate, which can lead to more complicated properties. This also lead to challenges to compare with any literature values (e.g., chemical composition, volatility, thermal properties, etc.)

We added SO$_2$ to promote particle formation (Stangl et al., 2019) by forming sulfuric acid (H$_2$SO$_4$) via the reaction between SO$_2$, OH, and O$_3$, but not to investigate S-containing organic species. Indeed, the mass fractions of S-containing organic species are very low (<5%) in all cases (SOA mass between 1 – 15.6 µg m$^{-3}$). Adding SO$_2$ prevents the use of very high VOC and O$_3$ concentrations due to the small residence time within the aerosol flow reactor to produce SOA. Therefore, the change of properties (e.g., chemical composition, volatility) of the SOA is expected to be limited in this study.

We modified the sentence in Line 251: *"SO$_2$ is injected from a commercial cylinder (500 ppm, AIR PRODUCTS Inc.), to promote the particle formation and to generate more SOA mass."*

2). I do not understand how you did sensitivity corrections. Please explain that a little bit more.

The key step of sensitivity correction is to determine the correlation between sensitivity and dV$_{50}$ (voltage difference at half signal maximum intensity) as proposed in Figure S8. This is obtained by using the standard compounds exhibiting different binding energy (i.e., sensitivity). Then, according to the function derived from Figure S8, the concentration of the different particle phase species can be estimated after measuring their dV$_{50}$ values from the CID scanning as described by Lopez-Hilfiker et al. (2016). We clarify the sensitivity correction in the following section

Lines 385 – 388: *"The correlation between sensitivity (Figure S2) and dV$_{50}$ (Figure S7) is obtained based on the standard compound, as depicted in Figure S8. By applying this sigmoidal function and using the dV$_{50}$ values determined for individual α-pinene-derived SOA products, the concentration of every oxidation compound can be estimated."*

3). It is very surprised to me that your SOA mass derived from CIMS is much higher than that from SMPS. Those results do not convince me since I expect the mass to be underestimated by CIMS due to evaporation and transport efficiency, re-condensation on the wall, and loss of volatile species in the activated carbon denuder in WALL-E. I do not really understand why WALL-E leads to overestimation.

The uncertainties pointed out by the reviewer are already included within the calibration factors derived from the introduction of the different standards (i.e., Figures S2 and S7-8). The overestimation is not related to the WALL-E interface but to the uncertainties related to the declustering method as discussed in previous work (e.g., Lopez-Hilfiker et al., 2016). For example, the presence of different isomers can lead to very different calibration factors (Lee et al., 2014). We would like to point out that the quantification obtained for sulfuric acid is within very good agreement (i.e., 7% using 8-compound fitting) between the direct calibration (i.e., Figure S2 and the concentration retrieved by the declustering scan method. While suffering from some uncertainties, the declustering scan represents a unique opportunity to provide a semi-quantification of all compounds present in the particle phase for untargeted analysis. We respectfully disagree with Reviewer #2 that the SOA mass derived from the chemical characterization is "much higher" than from SMPS. Uncertainties related to CIMS quantification have often been reported to be 50-100%. A dedicated work will be required to narrow down the uncertainties by notably using non-commercial standards (e.g., Gagan et al., 2023; Kenseth et al., 2023), which is outside the scope of this study.

3. I do not find discussion about volatility based on their measurements.

In this work, we focus on characterizing the system performance and introducing the capability to estimate volatility. A more detailed and comprehensive volatility analysis, particularly for relevant SOA systems, will be addressed in future studies.

4. Are there any size dependencies in your results?

This has been previously discussed in Zhao et al. (2024), who showed that the evaporation within the TD is not size-dependent at temperatures greater than 300°C. All experiments

performed in this work (besides the temperature ramp) were performed at 320°C, so we do not expect any size dependencies in the results presented here.

5. What is the sensitivity and detection limitation of using WALL-E?

The corresponding sensitivity of α-pinene-derived SOA compounds generally increases with molecular mass and reaches a plateau corresponding to the maximum sensitivity (i.e., collision limit), as shown in Figure S10. This indicates an upper limit of sensitivity of 0.08 ncps·per·μg·m$^{-3}$, yielding a detection limit of about ~5-10 pg·m$^{-3}$ (assuming a detection limit with the CIMS of $5 \times 10^{-6} - 1 \times 10^{-5}$ as discussed in Riva et al. (2020)).

We revised lines 393 - 395: *"with an upper limit of sensitivity of 0.08 ncps·per·μg·m$^{-3}$ providing a limit of quantification ~5-10 pg·m$^{-3}$ corresponding to uncorrected signal intensities of $5 \times 10^{-6} - 1 \times 10^{-5}$ (Riva et al., 2020)."*

Specific comments:

1. could you label each part and the flow direction? I also do not understand why there are two tees for the sheath flow. Is WALL-E like a distillation tube where a sample tube is inserted inside a big tube? Then why are these tees in the same position?

As mentioned in comment #4 of reviewer #1, Figure 1 has been revised to provide a clearer visual understanding of the setup

2. Section 2.2.1. It is unclear to me how you mixed these solutions. What is the fraction of each chemical?

We mixed these standard chemicals in water, with the concentration of each to be 1 ppm. To clarify, we add a sentence:

Lines 222 – 223: *"The concentration of each chemical in the aqueous solution is 1 ppm."*

3. L330-331, "Without sensitivity … ug m-3." The signal is only 0.04 ncps, which is the same as the signal that no VOC was injected (L326). Therefore, the SOA signal could just be the background noise.

The mass spectra presented in this work (including Figure 4 and lines 330-331) are background-subtracted. The total signal of all detected SOA compounds is ~0.08 ncps (without background subtraction) and ~0.04 ncps (after background subtraction) when the particle mass is $1.0 \pm 0.1$ μg m$^{-3}$. However, the high background of 0.04 ncps in line 326 refers to the individual compound $C_8H_{12}O_4$, which could be a contamination compound from the flow tube. Therefore, the SOA signal must not be the background noise.

4. Figure S6. I suggest using scattering plots with fittings.

Figure S6 is now in scattering plots with fittings.

5. Figure numbers in SI need to be correct.

They are now corrected.

6. dV50 is not well defined.

The definition is now added.

7. L387-389, "Using the dV50 … 8-compound fitting)." Do you expect that high amount of H2SO4? How much SO2 was added to the system?

Yes, we expect a high amount of $H_2SO_4$ of 6.4 – 8.7 µg m$^{-3}$ in the case of SOA mass of 15.6 µg m$^{-3}$. $SO_2$ concentration is 31 ppb as reported in Table S3.

8. Section 3.3.2. I think either $T_{max}$ or $T_{50}$ works for discussing the volatility, but you are mixing them up and making it hard for me to follow. I suggest either picking one or separating them into two sections.

In this work, we discuss both $T_{max}$ and $T_{50}$ to provide continuity with previous studies while also introducing the benefits of using $T_{50}$. To improve clarity, we have revised this section to separate the discussion of each parameter better and indicate when $T_{max}$ is used for comparison purposes and when $T_{50}$ is applied for volatility estimation.

Line 447: *"3.3.2 $T_{50}$ as a Robust Volatility Metric"*

Lines 490-493: *"Although $T_{50}$ is introduced as a more robust metric in this study, temperature correction and inter-comparison with other techniques have traditionally been performed using $T_{max}$ values. Therefore, we present the correction and comparison based on $T_{max}$ to ensure consistency with previous studies before applying the $T_{50}$ approach."*

Lines 517 – 520: *"3.3.4 Volatility Estimation from $T_{50}$*

*As previously explained, the use of $T_{50}$ would provide a more reliable estimation of the volatility, in the case of WALL-E, which is typically inferred using the relationship between $T_{50}$ and the saturation concentration ($C^*$) as discussed in prior studies (Ylisirniö et al., 2021)."*

9. L454-455, "For comparison … of 98%." Why do you use 98%, not 99.5% or 99.9%, as other studies you mentioned before?

In prior studies, $T_{max}$ was straightforward to determine due to Gaussian-shaped thermograms. In our case, the WALL-E thermograms follow a more sigmoidal shape, making $T_{max}$ less robust and highly sensitive to the selected threshold. To demonstrate this, we conducted a sensitivity analysis (Figure S12), which shows that while the absolute $T_{max}$ values vary slightly with the threshold (e.g., 98% vs. 99.9%), the trend and slope of the volatility comparison remain unchanged. We selected 98% because it better aligns with the observed signal maxima, offering more representative $T_{max}$ values for comparison purposes.

Lines 470-473: *"In contrast to earlier systems, including the FIGAERO and VIA, which often produce Gaussian-shaped thermograms with clearly defined peaks, WALL-E thermograms follow a sigmoidal profile. This makes the determination of $T_{max}$ more sensitive to the selected threshold and prone to variability, especially at upper temperature ranges."*

10. L457-458, "In addition … fewer data points." Do you mean fewer data points at higher temperatures? Overall, you have lots of data points, and it seems that only PEG-17 do not have enough data points after reaching the $T_{max}$.

This limitation mainly affects PEG-17, while sufficient data points are available for the other PEGs.

Lines 479-481: *"In addition, at higher temperatures, the fit for less volatile PEGs can result in more errors due to fewer data points, which mainly affect PEG-17. Taking a threshold…"*

11. L460-463, "The signals … TD techniques." How much improvement compared to other techniques? Did other techniques also use PEG?

While an exact numerical comparison is not available, the improvement is demonstrated by the thermograms following a sigmoidal shape rather than a Gaussian shape, indicating smoother evaporation. PEG standards were also used with VIA, where steeper decreases beyond $T_{max}$ were observed, suggesting higher fragmentation or losses within the interface (Zhao et al., 2024).

12. Figure S14, please provide references for the FIGAERO data.

The reference is added

13. L485-486, "Longer residence … values." I think this should be the opposite. Shorter residence time particles might not reach thermal equilibrium, so they need a higher temperature to evaporate completely.

We would like to thank the reviewer for pointing out this error. The plot labels were corrected, and the text was revised. 60 ms represents the residence time with a sample flow rate of 0.75 SLPM and hot flow rate of 0.25 SLPM, while 72 ms represents the time with 1 SLPM sample flow rate and 0.25 SLPM hot flow rate.

Lines 513 – 516: *"Longer residence times allow for gradual heating and equilibration, leading to lower temperature values. In contrast, shorter residence times accelerate desorption, resulting in higher temperature values due to insufficient thermal equilibration."*

SI Figure S15: *"Figure S15. Effect of the residence time inside the TD region on the $T_{max}$ corrected $T_{50}$ values for PEG. 72ms represents a sample flow rate of 0.75 SLPM and HF of 0.25 SLPM, while 60ms represents SF of 1 SLPM and HF of 0.25 SLPM."*

14. Figure 7: I don't understand the purpose of this figure since you did not show any of your data.

Figure 7 presents the $T_{50}$ values determined using WALL-E and illustrates how different volatility estimation methods lead to varying predictions. This framework helps visualize the uncertainty across methods and, based on the highlighted regions, allows us to estimate the volatility class a measured compound would fall into.

15. Figure 8: Is the mean line showing the average of heat and cool? It is not clear to me how you used the data from the fast cool ramp.

We applied the sigmoid fitting method to both the heating, cooling, and fast cooling ramps, extracted the $T_{50}$ values from each, and averaged them. The orange line represents the averaged $T_{50}$ value.

Lines 549-554: *"Figure 8: Comparison of estimated $T_{50}$ values for α-pinene derived SOA compounds, specifically $C_{10}H_{16}O_{6-9}$ with volatility predictions from COSMO-RS and SIMPOL models. The measured $T_{50}$ values (orange) are shown with their min/max range, while the shaded regions represent model predictions taking into consideration the isomerisation. We*

*applied the sigmoid fitting method to both the heating, cooling, and fast cooling ramps, extracted the $T_{50}$ values from each, and averaged them. The orange line represents this averaged $T_{50}$ value."*

16. L535-537, "By using … 1 ug m-3)." I am not fully getting this. Where did you show these results?

Concentrations were estimated based on the declustering method discussed in this work. Hence, we determined the concentration of individual SOA compounds. As shown in Figure R1, the products present in α-pinene-derived SOA range from ~10 pg m$^{-3}$ to 70 ng m$^{-3}$ for a total SOA mass of 1 µg·m$^{-3}$ measured by the SMPS.

[Figure]

Figure R1. Product concentrations of individual α-pinene-derived SOA compounds were quantified using the declustering procedure using the 8- and 10-compound calibration solution. Total SOA mass of 1 µg m$^{-3}$ was measured by an SMPS.

---

## Author Comment (AC2)

The authors thank the reviewer for the careful review of our manuscript and the helpful comments and suggestions. All the comments (in black) are addressed point by point, with our response in blue, and the corresponding revisions to the manuscript in red.

**Review # 1**

This article provides a detailed review of existing methods and introduces a new analytical method called WALL-E, which is designed to measure the chemical composition of atmospheric particles in real time. The work is significant and innovative, but the readability of the article needs improvement. Here are the specific details:

1. Introduction: The Introduction compares many methods, but it doesn't clearly state that the main contribution is adding a device that integrates TD (thermal desorption) and other functions in front of the Br-CIMS. This might be confusing for users of the instrument, especially those who are not developers, as they may struggle to quickly understand the research goal. In addition, the introduction of VIA-NO₃-CIMS is too simplistic and needs to be strengthened.

We added and modified the last paragraph of the introduction to clarify the research goal of this manuscript:

Lines 106-109: *"Currently, there is no technique based on thermal evaporation able to prevent thermal fragmentations, and suitable for the on-line measurement of moderate oxygenated (e.g., molecular oxygen atoms <6) organic species."*

Lines 123 – 126: *"In this work, we designed a newly designed wall-free particle evaporator (WALL-E) to perform online organic particle characterization while preventing ionization-induced fragmentation and minimizing thermal decomposition effects. WALL-E is coupled to a chemical ionization inlet attached to a CIMS to achieve real-time measurements of aerosol particles at a molecular level."*

We also strengthened the introduction of VIA-NO3-CIMS:

Lines 98-103: *"…, which mainly consists of a sulfinert-coated stainless steel as TD unit and a following cold dilution flow of $N_2$. The evaporation tube of VIA is bonded with an insert silica layer into the surface; a dilution flow is used to cool down the sampling flow and minimize the recondensation of the evaporated compounds. The parameters of the dilution unit are critical factors that affect the final sensitivity of the entire system."*

2. Abstract: The phrase "suffers from different artifacts" in the Abstract is too vague. It should specify what kind of artifacts WALL-E addresses.

We clarify it by mentioning specific artifacts.

Line 25-26: *"suffers from different artifacts (i.e., thermal decomposition, fragmentation, wall loss)."*

3. Undefined Term in Abstract: $T_{50}$ is not defined in the Abstract.

$T_{50}$ is now defined in the Abstract.

Line 40-41: *"In addition, the measured $T_{50}$ (the temperature at which 50% of a compound evaporates), for α-pinene-derived SOA ..."*

4. Figures 1 and 2: Combining Figures 1 and 2 would make it easier to see both the appearance of the TD and the airflow simulation inside. It would also be helpful to label the "Dilution/Cooling Unit" and "Gas-phase Denuder."

We appreciate the reviewer's suggestion. In response, we have revised Figure 1 to include a new panel (Figure 1B) that clearly defines all key regions of the system, including the "Dilution/Cooling Unit" and the "Gas-phase Denuder," as well as the direction and entry points of the different flows. This provides a clearer visual understanding of the setup and complements the airflow simulation shown in Figure 2.

Lines 144-149:

[Figure]

*Figure 1: (A) Design of the WALL-E interface. (B) Schematic of WALL-E with a gas-phase denuder (GPD) connected to the inlet. The thermal desorption region (TD) is shown in red, where the hot flow (HF) is mixed with the sample flow (SF). A ceramic spacer is indicated by the dashed rectangle. The cooling region (CR), shown in blue, is where the cooling flow (CF) is introduced.*

5. Lines 346-363: While it's possible to guess what DC and $dV_{50}$ mean, it would be clearer if their full names were provided.

The full names of DC (direct current) and $dV_{50}$ (voltage difference at half signal maximum intensity) are now added to the main text.

Lines 370 – 375: *"By utilizing the in-source collision ion dissociation feature (Riva et al., 2019), which corresponds to an increase in the direct current (DC) offset voltages between two ion optics within the flatapole, the binding energy of the [M-Br⁻] adducts can be probed (Figure*

*S7). The voltage difference at half signal maximum intensity (dV$_{50}$) broadly ranges from 5.1 to 20.2 Volts, indicating differences in binding energies and varying clustering strengths to Br$^-$."*

6. Lines 377-381: The explanation here is not clear or intuitive enough in the following.

The sentence is rewritten.

Line 402-406: *"Among all calibrated compounds, two outliers exist (i.e., shikimic acid and glucose), which might be due to partial evaporation leading to an underestimation of the sensitivity. When excluding these two species, the estimate of total particle mass concentration is closer (with 36% overestimation) to the total particle mass concentration measured by the SMPS as depicted in Figure 5."*

7. Lines 407-418, Lines 430-438: This part repeats information from the Introduction. It should be shortened and focus on the core issues. Some part is better for Introduction

To improve readability and reduce redundancy, we have made the following changes:

A shorter version of the general discussion on volatility determination was added to better introduce the challenges and context of our work:

Lines 119-122: *"Additionally, for highly oxygenated and multifunctional compounds, volatility determination remains particularly uncertain, as isomerism and intermolecular interactions can significantly influence evaporation behavior (Lee et al., 2014; Bannan et al., 2019)."*

Section 3.3.1 (Thermograms and $T_{max}$ Determination):

The general discussion on volatility determination was removed to avoid repeating content already presented in the introduction. This section now focuses directly on the experimental results and the specific improvements provided by WALL-E.

Lines 433-446: *"The determination of volatility represents one of the greatest analytical challenges when characterizing aerosol particles, as it depends on multiple factors, including molecular composition, intermolecular interactions, and experimental conditions (Compernolle et al., 2011). Various experimental and theoretical techniques have been developed over the last decades to retrieve volatility information, each with advantages and limitations.*

*While FIGAERO, VIA have been widely used, their design constraints introduce inherent limitations, can introduce artifacts such as recondensation, analyte interactions, and fragmentation. The prolonged residence time on the FIGAERO may also lead to early desorption of volatile species or chemical reactions between co-deposited compounds, impacting the accuracy of volatility estimates (Stark et al., 2017; Schobesberger et al., 2018; Buchholz et al., 2020). VIA thermograms, on the other hand, show evidence of fragmentation and thermal decomposition at high temperatures (Zhao et al., 2024b). WALL-E introduces a new approach, optimizing the balance between thermal residence time and evaporation efficiency, allowing for precise volatility determination with reduced wall interactions."*

Section 3.3.2 (From $T_{max}$ to $T_{50}$):

The Title was changed

Line 447: "$T_{50}$ as a Robust Volatility Metric"

No changes were made in the paragraph. We believe this section is essential to clearly explain and justify the introduction of $T_{50}$ as a key innovation in this work.

8. Line 474: The statement "The corrected values align well with those reported for VIA and FIGAERO" is too general. It should include references and specific results.

The corresponding values and references for the VIA and FIGAERO data are provided in Figure S14 of the Supplementary Information, the sentence was modified as follow:

Lines 500-503: *"The corrected values align well with those reported for VIA and FIGAERO, maintaining a consistent trend across different molecular weights (Figure S14)."*

9. Section 3.3: The main innovation is the use of $T_{50}$, but the subsection titles are confusing. $T_{max}$ is discussed in Sections 3.3.1 and 3.3.3, while $T_{50}$ is in Sections 3.3.2 and 3.3.4. This arrangement makes it hard for readers to follow.

To clarify the structure and maintain consistency with previous studies, we have added a transition sentence at the beginning of the Temperature Correction and Comparison section. This explains why $T_{max}$ is used for the correction and inter-comparison before introducing the improvements provided by $T_{50}$.

Lines 490-493: *"Although $T_{50}$ is introduced as a more robust metric in this study, temperature correction and inter-comparison with other techniques have traditionally been performed using $T_{max}$ values. Therefore, we present the correction and comparison based on $T_{max}$ to ensure consistency with previous studies before applying the $T_{50}$ approach."*

---

## Author Comment (AC3)

The authors thank the reviewer for the careful review of our manuscript and the helpful comments and suggestions. All the comments (in black) are addressed point by point, with our response in blue, and the corresponding revisions to the manuscript in red.

**Review # 3**

Gao et al. presented a novel technique WALL-E, which is a thermal desorption unit coupled with CIMS. It can detect and quantify the chemical composition of aerosol particles with CIMS in real-time. This study is very interesting and innovative. However, the manuscript's readability could be improved. I recommend a major revision prior to publication. Please see my detailed comments below:

General comments:

1. Stainless steel tubing was used for WALL-E in this study. Uncoated stainless steel tubing can adsorb semi-volatile and polar organics at elevated temperature, leading to sample loss and memory effects. Did you use any inert-coating on these tubes? if not, I recommend using inert-coated tubing (e.g. sulfinert coated stainless steel tubing) in the future, and add discussion of the caveat of using uncoated stainless steel tubing

The system was not coated during the experiments presented in this study, but coating will be implemented in future developments to further reduce potential wall interactions

Lines 171-174: *"A limitation of the current design is the use of uncoated stainless steel, which can lead to adsorption or memory effects for more-volatile compounds. Inert coatings will be considered in future iterations to further minimize wall interactions."*

2. SMPS was used in the experiments, and I wonder if the evaporation efficiency, $T_{50}$, and volatility characterization are particle size and mass loading dependent? Please include the number and size distribution for standards and SOA. Are SMPS and CIMS measure the same particles?

This has been previously discussed in Zhao et al. (2024), who showed that the evaporation within the TD is not size-dependent at temperatures greater than 300 °C. All experiments performed in this work (besides the temperature ramp) were performed at 320 °C, so we do not expect any size dependencies in the results presented here.

Chemical characterization of α-pinene-derived SOA, the SMPS, and the WALL-E system were positioned at the same distance, using the same tubing material and having the same sampling flow. As a result, both instruments measured the same particles.

3. There are two figure S5 and two figure S6 in the SI. It's very hard to follow which plot is being referred to. Please correct.

They are now corrected.

4. For the sensitivity calibration, the unit of ncps/(ug/m$^3$) was used. When comparing the sensitivity among different compounds and calibration (second figure S6 and figure S8), the unit of ncps/ppm should be used to eliminate the influence of molecular weight.

Because this is a particle phase measurement, the unit ug/m³ is determined based on the averaged expected particle density (i.e., reported density for a pure product) and not a gas phase concentration. As a result, the use of ncps/ppm is not appropriate in this case.

5. In section 3.3.2, the usage of $T_{max}$ and $T_{50}$ is confusing. The $T_{max}$ in FIGAERO usually refer to the temperature at which the signal intensity is maximum. For VIA, people usually use $T_{50}$ as well instead of $T_{max}$, as VIA thermogram also shows a sigmoid curve instead of a near-Gaussian shape. The WALL-E and VIA both have continuous aerosol flow into the TD, therefore they have similar thermograms. The $T_{max}$ for FIGAERO and $T_{50}$ for VIA and WALL-E both represent the temperature when the desorption rate is maximum.

We agree that $T_{50}$ is more appropriate for WALL-E due to its sigmoidal thermograms, and it is the main metric used for volatility analysis in this work. $T_{max}$ is included only for comparison with other systems where it has traditionally been used. While $T_{50}$ was used in one VIA study to describe volatility trends across PMF factors (Li et al., 2023), most VIA studies, including the PEG-based calibration, use $T_{max}$ to derive volatility (C*) (Zhao et al., 2024). Regarding thermogram shapes, thermal decomposition can result in steep decreases after $T_{max}$, which is not observed with WALL-E, where a near-sigmoid profile is obtained

6. Are all the standards and SOA particles fully evaporated after passing through the WALL-E?

The mass remaining in the particles after the evaporation in WALL-E is dependent on the sample flow and particle mass (Figure R2). Therefore, according to the flow used in this work for SOA experiments, we expect that > 94% particles were evaporated.

[Figure]

Figure R2. The fraction of evaporated particles corresponding to the particle number concentration under different sample flows (0.3-1.5 SLPM).

For the standards, reported in Figure S8, the sensitivity of shikimic acid and glucose are outlier compounds which the sensitivities are lower than expected (the fitted sigmoidal curve). This

may indicate the partial evaporation of those low-volatile species or an error in the averaged density.

We added a sentence.

Line 402-406: *"Among all calibrated compounds, two outliers exist (i.e., shikimic acid and glucose), which might be due to partial evaporation leading to an underestimation of the sensitivity. When excluding these two species, the estimate of total particle mass concentration is closer (with 36% overestimation) to the total particle mass concentration measured by the SMPS as depicted in Figure 5."*

Specific comments:

1. Line 38: $T_{50}$ is not defined in the abstract.

It's now added.

Lines 40 – 41: *"In addition, the measured $T_{50}$ (the temperature at which 50% of a compound evaporates), for…"*

2. Line 157: what is the residence time in the TD with sample flow rate at 1 SLPM?

The residence time at 1 SLPM flow is between 70-90 ms, depending on the hot flow (0 - 0.5 SLPM).

Lines 178 – 180: *"At a sample flow rate of 1 SLPM, the residence time in the TD is estimated to be between 70 and 90 ms, depending on the added hot dilution flow (0 to 0.5 SLPM)."*

3. Line 217: what are the densities used for the mass concentration calculated by SMPS for each compounds? Consider include into table S2.

The density for each compound was added to Table S1.

4. Line 233: how is OH radical generated in the OFR? Please include the ozone concentration, relative humidity, and OH exposure in the table S3, as $O_3$/OH initiated oxidation was mentioned.

OH radicals are generated in the reaction of VOC + $O_3$. As we did not add any OH scavenger, the yield of OH radicals is expected to be ~85%, which has been reported by other studies (Rickard et al., 1999; Paulson, 1998). Unfortunately, the $O_3$ concentration was not measured. We added relative humidity in Table S3.

5. Line 294: any results indicating the better flow stability with the presence of a HF?

As shown in Figures 3C and D, the addition of the HF improves the evaporation and the transfer of the compounds to the dilution unit.

6. Line 317-318: I wonder if you observe any trimers? How about the sulfur containing compounds shown in figure 5B, as $SO_2$ was added into the system.

No, we did not detect trimer signals as their masses might be outside the measurement range of the CIMS used (i.e., up to m/Q 700). The mass fraction of S-containing compounds is <5%.

7. Line 326: what is ncps? Normalized count per seconds? How are they normalized?

We added the definition and clarify the normalization.

Line 280: *"All product ions are normalized to the Br⁻ signals."*

Line 347 - 348: *"It has an unexpected high background of ~0.04 normalized counts per second (ncps) when there is no VOC injected."*

8. Line 330-331: "without sensitivity correction, …, mass is $1.0 \pm 0.1$ μg/m³" 0.04 ncps is the same as the background as mentioned in line 326. I'm not sure if I fully understand this sentence here.

The mass spectra presented in this work (including Figure 4 and lines 330-331) are background-subtracted. The total signal of all detected SOA compounds is ~0.08 ncps (without background subtraction) and ~0.04 ncps (after background subtraction) when the particle mass is $1.0 \pm 0.1$ μg m-3. However, the high background of 0.04 ncps in line 326 refers to the individual compound $C_8H_{12}O_4$, which could be a contamination compound from the flow tube. Therefore, the SOA signal must not be the background noise.

9. Line 356: Figure S8, why shikimic acid and glucose were excluded? Any criteria?

As shown in Figure S8, shikimic acid and glucose were outliers from the fitted sigmoidal curve. As discussed in previous studies, selected standard compounds might induce uncertainty in the sensitivity estimations (Zaytsev et al., 2019; Bi et al., 2021; Song et al., 2024). Therefore, to get a better calibration curve, we presented and discussed the results obtained with and without the outliners to evaluate the uncertainties associated with the approach.

One potential explanation for the lower sensitivity of shikimic acid and glucose is the partial evaporation of the compounds. We have added a sentence regarding the potential reason.

Line 402-406: *"Among all calibrated compounds, two outliers exist (i.e., shikimic acid and glucose), which are most likely due to partial evaporation leading to an underestimation of the sensitivity. When excluding these two species, the estimate of total particle mass concentration is closer (with 36% overestimation) to the total particle mass concentration measured by the SMPS as depicted in Figure 5."*

10. Line 388-390: how is sulfuric acid formed in the OFR? The particle phase $H_2SO_4$ mass loading seems to be pretty high. Any further discussion regarding the $H_2SO_4$? "a good agreement is retrieved" I'm not sure if I fully understand this. Were you saying the SMPS results match the CIMS results? But how can SPMS differentiate organic and inorganic? The mass loading reported with 10 compounds fitting (8.7 μg/m³) is higher than that with 8 compounds fitting (6.4 μg/m³), but the 10-compounds lead to underestimation, and 8 compounds lead to overestimation. Please correct.

Since $SO_2$ was added to promote the particle formation and to generate more SOA mass (Stangl et al., 2019), sulfuric acid ($H_2SO_4$) was formed via the reaction between $SO_2$ and OH (and $O_3$ and Criegee radicals), which is a very well-known chemical process leading to sulfuric acid.

The concentrations of sulfuric acid obtained from the direct calibration (i.e., Figure S2) and the declustering method were compared, providing an opportunity to evaluate the declustering method.

We correct the 'underestimation' and 'overestimation' in the sentence as follows:

Lines 414 - 417: *"A good agreement between the direct calibration (Table S2) and the declustering scan method is retrieved (7% overestimation and 25% underestimation for the 10 and 8-compounds fit, respectively underlining the benefit of using this approach to obtain the concentrations of organic and inorganic present in the particles."*

11. Line 454-455: as mentioned in the general comment 4, I'm not sure if you should use $T_{max}$ from WALL-E to compare with other particle evaporators, or use $T_{50}$, which is the temperature when the desorption rate is maximum.

In this case, using either $T_{max}$ or $T_{50}$ would not make a significant difference for what we are trying to demonstrate, as both values have similar slopes and fall between the VIA and FIGAERO results. $T_{max}$ is used for comparison with prior studies. The following plot includes the added $T_{50}$ values for reference.

[Figure]

$T_{50}$ will be adopted as the primary metric for volatility characterization in future WALL-E studies.

12. Line 461-463: any explanation for the signal decrease after they reach the plateau? Also why this is an improvement compared to other online TD techniques?

We attribute the signal decrease after reaching the plateau to thermal decomposition or additional losses occurring beyond $T_{max}$. In other systems, this decrease is typically much steeper and does not follow a sigmoidal behavior; instead, the signal often drops rapidly, resembling a Gaussian shape. The improvement in our system is the ability to maintain a near-sigmoidal thermogram, indicating smoother evaporation and net reduction of the fragmentation compared to other online TD techniques.

13. Line 479-487: when you mention the $T_{max}$ for VIA, do you mean the gas temperature or measured temperature? As far as I know, VIA does not measure the gas temperature directly.

The $T_{max}$ reported in previous VIA studies represents the set value of the TD part, not the direct measurement of the gas.

14. Figure S1: chemical ionization section is not mentioned in the plot.

The plot was modified, and CI was added to the instrument box

15. Figure S4: for the lower plot, what does the time in x-axis mean? What different conditions are they corresponding to?

The time on the x-axis corresponds to the continuous temperature ramping process, not to averaged values at each set temperature. This plot shows how closely the signal remained around zero throughout the entire heating and cooling cycle from 0 °C to 390 °C and back to 0 °C.

16. Figure S12: please specify the legend. Also from the main text, I understand the reason to include the comparison between 99.5% and 98%. But why include 40% and 60%? $T_{50}$ should be from your sigmoid fitting results as mentioned in the main text.

The 40% and 60% thresholds were included to demonstrate that even if the fit is not ideal—which is not the goal but can occur in practice—the resulting error in volatility estimation remains negligible. The 10% variation was intentionally exaggerated to illustrate the limited impact of such deviations.

17. Figure S15: do you mean $T_{max}$ or $T_{50}$? $T_{50}$ in the plot, but $T_{max}$ in the caption and main text.

It's corrected now.